# Systemic LPS Administration Stimulates the Activation of Non-Neuronal Cells in an Experimental Model of Spinal Muscular Atrophy

**DOI:** 10.3390/cells13090785

**Published:** 2024-05-04

**Authors:** Eleni Karafoulidou, Evangelia Kesidou, Paschalis Theotokis, Chrystalla Konstantinou, Maria-Konstantina Nella, Iliana Michailidou, Olga Touloumi, Eleni Polyzoidou, Ilias Salamotas, Ofira Einstein, Athanasios Chatzisotiriou, Marina-Kleopatra Boziki, Nikolaos Grigoriadis

**Affiliations:** 1Laboratory of Experimental Neurology and Neuroimmunology, 2nd Neurological University Department, AHEPA General Hospital of Thessaloniki, Faculty of Health Science, Aristotle University of Thessaloniki, 54636 Thessaloniki, Greece; elenikarafoulidou95@hotmail.com (E.K.); bioevangelia@yahoo.gr (E.K.); ptheotokis@gmail.com (P.T.); chrystallakon@outlook.com (C.K.); nellamaria913@gmail.com (M.-K.N.); ilianamihaelidou@hotmail.com (I.M.); toulolga@auth.gr (O.T.); elpolyz@auth.gr (E.P.); iliassalamotas@gmail.com (I.S.); 2Department of Physical Therapy, Faculty of Health Sciences, Ariel University, Ariel 40700, Israel; ofirae@ariel.ac.il; 3Department of Physiology, Medical School, Aristotle University of Thessaloniki, 54636 Thessaloniki, Greece; chatziso@auth.gr

**Keywords:** spinal muscular atrophy, SMNΔ7, microglia, astrocytes, gut–brain axis, gut–skeletal muscle axis

## Abstract

Spinal muscular atrophy (SMA) is a neurodegenerative disease caused by deficiency of the survival motor neuron (SMN) protein. Although SMA is a genetic disease, environmental factors contribute to disease progression. Common pathogen components such as lipopolysaccharides (LPS) are considered significant contributors to inflammation and have been associated with muscle atrophy, which is considered a hallmark of SMA. In this study, we used the SMNΔ7 experimental mouse model of SMA to scrutinize the effect of systemic LPS administration, a strong pro-inflammatory stimulus, on disease outcome. Systemic LPS administration promoted a reduction in SMN expression levels in CNS, peripheral lymphoid organs, and skeletal muscles. Moreover, peripheral tissues were more vulnerable to LPS-induced damage compared to CNS tissues. Furthermore, systemic LPS administration resulted in a profound increase in microglia and astrocytes with reactive phenotypes in the CNS of SMNΔ7 mice. In conclusion, we hereby show for the first time that systemic LPS administration, although it may not precipitate alterations in terms of deficits of motor functions in a mouse model of SMA, it may, however, lead to a reduction in the SMN protein expression levels in the skeletal muscles and the CNS, thus promoting synapse damage and glial cells’ reactive phenotype.

## 1. Introduction

Spinal muscular atrophy (SMA) is a consequence of homozygous loss of *SMN1* expression and insufficient expression of functional SMN from the *SMN2* gene. SMA is inherited in an autosomal recessive pattern in the majority of cases [1]. A hallmark of the disease is the progressive degeneration of the lower alpha (α) motor neurons (α-MNs), which are located within the anterior horn of the spinal cord. Degeneration of the a-MNs leads to severe motor impairment and generalized muscle atrophy. SMN protein (UniProtKB-Q16637/SMN_HUMAN) is pivotal for the proper function of all living cells and is essential for a plethora of cellular processes, such as the maturation of precursor mRNAs (pre-mRNA) [2]. Recently available disease-modifying therapies have altered the classical paradigm of genetically determined disease phenotypes. Categorization based on patient walking ability (non-sitters, sitters, and walkers) has been advocated, thus underlining the potential of the aforementioned treatments to benefit patients in terms of overall function and survival [3]. Environmental factors and their potential impacts on pathophysiological processes in SMA have been studied. 

In SMA, the gut–muscle and the gut–brain axes have emerged as crosstalk pathways capable of altering disease characteristics [4]. Muscle atrophy in mice has been associated with microbial antigens, such as the Gram-negative bacterial lipopolysaccharides (LPS), recognized by Toll-like receptor 4 (TLR4) on immune cells, and with gut microbial composition alterations [5,6,7]. Several muscle characteristics, such as muscle mass, volume, growth, and strength, were shown to be affected by microbial antigens [8]. More specifically, LPS-induced skeletal muscle atrophy has been shown in both in vivo and in vitro experiments [9,10]. 

Moreover, LPS and other microbial antigens are known to trigger glial cell activation in the central nervous system (CNS) [11]. In motor neuron disease, both microglia and astrocytes contribute to the production of inflammatory cytokines and the infiltration of T-lymphocytes in the CNS [12]. Recent data point to the importance of microglial cells activation in amyotrophic lateral sclerosis (ALS), a rare motor neuron disease which shares clinical and pathological similarities with SMA [13,14]. Microglial cell and astrocyte activation has been reported earlier in SMA cases, although the exact role of these cell populations in SMA pathology remains controversial. Whether reactive glial cells promote inflammation in the context of disease pathogenesis or represent a bystander effect on damaged neurons in late-symptomatic stages remains elusive [15,16].

In this study, we aimed to address the effect of systemic microbial antigen administration on CNS innate immunity in experimental SMA. *E. coli* LPS was administered with intraperitoneal injections in the SMNΔ7 model of SMA, and the effect was assessed via clinical examination of the diseased mice, as well as by histological and molecular analyses of the skeletal muscle and CNS tissues.

## 2. Materials and Methods

### 2.1. Mice Colony and Animal Facilities

FVB.*Cg^Grm7Tg(SMN2)89Ahmb^Smn1^tm1Msd^Tg(SMN2*delta7)4299Ahmb/J* (Strain #005025) breeding couples were purchased from The Jackson Laboratory (Bar Harbor, ME, USA) in order to establish a colony of mice and acquire the SMNΔ7 mouse model of moderate Type II SMA. Mice were housed in a pathogen-free animal facility at the Laboratory of Experimental Neurology and Neuroimmunology of the 2nd Neurological University Department and the Laboratory of Development-Breeding of Animal Models and Biomedical Research at Faculty of Health Sciences of the Aristotle University of Thessaloniki (EL-54-BIOexp-10). All procedures were performed in accordance with the European Union Guidelines (Official Journal of the European Communities/No L 374/11) and the Greek Government Legislation. Monogamous mating was arranged between hemizygous *SMN2^+/+^;SMNΔ7^+/+^;Smn^+/−^* adult mice in order to obtain offspring of both the *SMN2^+/+^;SMNΔ7^+/+^;Smn^+/+^* (Control) and *SMN2^+/+^;SMNΔ7^+/+^;Smn^−/−^* (SMNΔ7) genotypes. 

### 2.2. Mouse Genotyping

DNA extraction was performed with a DNA nucleotype mouse PCR direct kit for genotyping (Macherey–Nagel, Düren, Germany) according to the manufacturer’s protocol. The PCR step was performed with a KAPA Taq DNA Polymerase kit (Kapa Biosystems, Wilmington, MA, USA) on a PCR thermal cycler (Biorad, Hercules, CA, USA) under the recommended three-primer system for Standard PCR Assay—Smn1<tm1Msd> (Protocol 22761, Version 2.3) by The Jackson Laboratory. The exact sequences from 5′ to 3′ primers used for the three-primer system were: common/forward—CTC CGG GAT ATT GGG ATT G, SMNΔ7; reverse—GGT AAC GCC AGG GTT TTC C; and wild-type reverse—TTT CTT CTG GCT GTG CCT TT. Electrophoresis for PCR products on 2% agarose gel was performed using a Mupid-one electrophoresis system with a FastGene ladder marker (Nippon Genetics, Düren, Germany). The PCR product of each reaction was visualized with a FastGene Fas-Nano system (Nippon Genetics, Düren) and was used to predict the genotype. Mice which were homozygous for the presence of endogenous Smn1^tm1Msd^ gene and having the *SMN2^+/+^;SMNΔ7^+/+^;Smn^+/+^* (Control) genotype produced a single clear product of 800 bp. Instead, mice homozygous for the absence of endogenous Smn1^tm1Msd^ gene having the *SMN2^+/+^;SMNΔ7^+/+^;Smn^−/−^* (SMNΔ7) genotype gave a single product of 500 bp. Hemizygous mice, as expected, showed two distinct lanes, one at 500 bp and a second lane at 800 bp. 

### 2.3. LPS Administration

In order to study the implication of microbial antigens in SMNΔ7 progression, 5 μg *E. coli* LPS (O55:B5; Sigma-Aldrich, Burlington, MA, USA) in PBS or PBS alone (vehicle) was administered by intra-peritoneal injection on neonates at postnatal day 5 (P5) at the onset of the first symptom. 

### 2.4. Behavioral Analysis and Clinical Evaluation

A novel set of non-invasive assessments was chosen to detect muscular deficits [17]. All procedures assessed the positions of the legs and tail in combination with general posture, in order to detect neuromuscular impairment. Body weight was monitored daily after birth until the late-symptomatic stage of disease. Tail suspension tests were performed every other day from day P2 to the late-symptomatic stage in order to detect neuromuscular deficits. Each pup was suspended by the tail for 15s and acquired a score of 0 to 4 as follows: 0: hind limbs always close together; 1: hind limbs always close together in the absence of other signs of posture deficits; 2: hind limbs often close together; 3: hind limbs not completely apart; and 4: a normal-appearing state with hind limbs fully apart. The hind limb suspension test (or tube test) was used to measure the overall motor function and hind limb muscle strength. The test was conducted with each pup placed carefully on a plastic 50 mL centrifuge tube, forehead down, holding on only by the hind limbs. Pups were protected from injury in case of a fall by the placement of a cotton cushion. Hind limb score ranged from 0 to 4 as follows: 0: clasping of the hind limbs with tail down; 1: constant clasping of the hind limbs with raised tail; 2: hind limbs close to each other; 3: hind limbs are close together but rarely touching each other; and 4: normal-appearing hind limb separation with raised tail.

### 2.5. Tissue Harvesting and Processing

Euthanasia was conducted on day 13, which is considered the late symptomatic stage for the SMNΔ7 model. Following euthanasia with liquid isoflurane (Iso-Vet) for vapor inhalation, the whole brain, L1-L5 lumbar spinal cord, gastrocnemius and triceps brachii muscles, gut, and spleen were harvested for downstream procedures. For Western blot (WB) analysis, tissues were snap-frozen in liquid nitrogen and stored at −80 °C. For histological and immunohistochemical analyses, specimens were post-fixed overnight with cold 4% paraformaldehyde solution. Tissues were then processed and embedded in paraffin blocks according to standard procedures. Serial sections (6 μm) were cut with a microtome and placed on Superfrost (Fisher Scientific, Hampton, NH, USA) slides. 

### 2.6. Western Blotting

Snap-frozen tissues were homogenized in a standard lysis buffer solution on ice using a bead mill 4 homogenizer (Thermo Fischer Scientific, Waltham, MA, USA). Briefly, after determining the protein concentration using a DC protein assay kit (Biorad, California) for all tissues, 20–30 µg of protein lysate was subjected to SDS–PAGE electrophoresis and transferred onto a nitrocellulose or polyvinylidene difluoride (PVDF) membrane. A protein amount of approximately 20–30 µg has previously been used in studies on transgenic mouse models with protein depletion or intervention in the protein expression pattern [18,19,20] in order to better detect weak expression signals. For this reason, for CNS tissues, 20 µg of total protein lysate were loaded in PVDF membranes, whereas, for muscle tissues, 30 µg of protein lysates were used in nitrocellulose membranes. Actin, GAPDH, and tubulin are the main housekeeping proteins traditionally used for protein level normalization. For polysystemic diseases, such as SMA, the application of a single housekeeping protein is challenging due to the fact that the expression of housekeeping proteins varies significantly across tissues [21]. Beta-actin is a well-characterized protein commonly used as a housekeeping protein for SMA [18,22,23]. Moreover, developmental changes are associated with SMN deficiency in the SMNΔ7 model. Protein level comparison in the SMNΔ7 model poses an additional challenge due to the fact that housekeeping proteins exhibit variable expression levels [24,25]. In the present study, beta actin was used for CNS and GAPDH was used for muscle. Membranes were incubated with blocking solution of 5% non-fat milk for 1 h, and an overnight incubation with anti-SMN (610646, BD Transduction Laboratories, Franklin Lakes, NJ, USA), anti- beta actin (Cell signaling, 4970S, Danvers, MA, USA), or anti-GAPDH (Proteintech, 10494-1-AP, Rosemont, IL, USA) followed (Appendix A). The next day, membranes were washed with PBS supplemented with 0.1% Tween 20 (PBST) and incubated with the appropriate horseradish peroxidase (HRP)-conjugated secondary antibodies (Appendix A): anti-mouse IgG-HRP (Cell signaling, 7076S, Danvers, MA, USA) and anti-rabbit IgG-HRP (Sigma Aldrich, A0545, Burlington, MA, USA) for 1 h at room temperature (RT). Membranes were then washed with PBST and visualized with enhanced chemiluminescence solution (ECL, L00221, Lumisensor Genscript, Piscataway, NJ, USA). In order to reliably normalize protein levels, in the present study, protein density was initially determined by the use of a sensitive detection system (iBright™ FL1500 Imaging System, Thermo Fisher Scientific, Waltham, MA, USA). All protein levels were first normalized to the housekeeping protein (beta-actin or GAPDH) and then quantified in relation to control group (vehicle) to minimize protein loading variation. In order to ascertain the relative expression of the SMN protein, SMN expression for control (vehicle) mice was calculated and set as 1. Relative SMN expression for experimental groups was subsequently computed by dividing the normalized expression of each sample to the normalized expression in the controls (vehicle). This yielded a normalized, relative expression value for each of the samples analyzed. The results were quantified using ImageJ software. 

### 2.7. Histological Stainings

For Bielschowsky staining, silver nitrate was applied to sensitize sections for the following impregnation with an ammoniacal silver solution. Brainstem (gigandocellular reticular and intermediate reticular nucleus) [26] and spinal cord sections (L1–L5) were firstly immersed in xylene for deparaffinization and then gradually hydrated by submersion in 96–50% alcohol solutions. Sections were rinsed with distilled water and incubated in 20% silver nitrate solution. Sections were then washed with distilled water and incubated in silver ammonium solution. Afterwards, 0.2% ammonia solution and an acidic silver solution (developer) were applied. Sections were then incubated in 5% *w*/*v* sodium sulfate solution, thoroughly rinsed, and dehydrated. Finally, sections were incubated in n-butyl acetate solution and embedded with adhesive medium (DPX). Stained sections were observed under an οptical microscope (Zeiss Axioplan II). For the estimation of neurodegenerative deposition and axonal defects, quantification was performed in accordance with a semi-quantitative scale, and sections were graded according to the severity level as previously described [27,28,29]: 0: normal or even silver stain throughout the measured area; 1: small areas found to lack silver stain in at least one axon; 2: small but frequent areas that lack silver stain in at least one axon or apparent mild silver deposition; and 3: extensive loss of silver stain evident in at least one axon or severe silver deposition.

For Nissl staining, paraffin sections were deparaffinized in xylene, rehydrated in alcohol baths, and washed with tap water. Sections were then incubated in 0.1% cresyl violet solution (Abcam, Waltham, Boston, MA, USA) at 37 °C for improved penetration and stain enhancement. Sections were rinsed with tap water and alcohol solutions. Finally, they were mounted with Entellan medium and coverslipped. For MN count, healthy Nissl^+^ neurons with areas above 200 μm^2^ and clearly visible nuclei were counted in serial sections [30,31]. 

### 2.8. Immunohistochemistry Stainings

Brainstem (gigandocellular reticular and intermediate reticular nucleus) [26] and spinal cord sections were deparaffinized in xylene, hydrated in alcohol baths, and incubated in 0.3% hydrogen diluted in methanol. Antigen retrieval was performed in a steamer for 1 h. Sections were then incubated in blocking in 10% FBS (fetal bovine serum) and 2% NGS (normal goat serum) to prevent the non-specific binding. Sections were incubated with the appropriate primary antibodies overnight. The primary antibodies (Appendix A) used were the polyclonal rabbit anti-Iba-1 (019-19741, Wako) and the polyclonal rabbit anti-GFAP (Z0334, Dako). The secondary antibody (Appendix A) was a biotinylated goat anti-rabbit (Vector, Navsari, Gujarat). Sections were incubated with avidin (Sigma) and developed with 3,3′-diaminobenzidine (DAB). Hematoxylin was used as a counterstain. Quantification of the results was presented as cells/mm^2^. Morphological alterations in microglia were studied utilizing a combination between the standard method of ramification index (Ac/Ap) as a ratio between the cell’s area (A_c_) and its projection area (A_p_) with automated variable count tools [32,33]. Photomicrographs from immunohistochemistry-prepared tissues were used for data analysis with the ImageJ “analyze skeleton” plugin and the “fractal analysis” (FracLac) plugin [33]. Overall, an increase in the ramification index was indicative of the enhanced reactivity of Iba-1^+^ and GFAP^+^ cells.

### 2.9. Immunofluorescence Stainings

The detection of changes in synaptophysin (SYP), innate immunity cells, and complement in CNS was performed using brainstem and spinal cord sections. Antigen retrieval was performed in a steamer for 1 h and sections were blocked with 10% FBS solution. Sections were then incubated overnight with primary antibodies (Appendix A). The antibodies used were mouse anti-SYP antibody (MO776, Dako), rabbit anti-Iba-1 (019-19741, Wako), rabbit anti-GFAP (Z0334, Dako), goat anti-CD206 (sc-34577, Santa cruz, Dallas, TX, USA), mouse anti-iNOS (sc-7271, Santa cruz, Dallas, Texas), goat anti-S100A10 (AF2377, Biotechne, R & D systems, Minneapolis, MN, USA), and the mouse anti-complement C3b/C3c (HM1065, Hycult Bioteck, Uden, The Netherlands) antibodies. An extra incubation step with goat anti-mouse Fab fragment (Jackson ImmunoResearch, WestGrove, PA, USA) was applied only when mouse antibodies were applied and due to the fact that mouse tissue was used. The secondary antibodies (Appendix A) used were the goat anti-rabbit 488A (20012, Biotium, Fremont, CA, USA), goat anti-mouse 555 (20030, Biotium, Fremont, California), and donkey anti-goat 555 (20039, Biotium, Fremont, California). Dapi was used for nuclear staining. Analysis and identification of synapse appositions was performed using the open-source ImageJ software after a manual input assignment. Detailed morphometric analysis of the total distribution of the SYP marker was performed individually after modification of the threshold in order to quantify the results. Data derived from immunohistochemical analyses were presented as positive cells per image area (cells/mm^2^). The sections were examined using a fluorescent microscope (Zeiss Axioplan II), and the data were analyzed using the ImageJ software. 

### 2.10. Quantification and Statistical Analysis

For all the aforementioned experiments, the total number of mice in each group for both independent experiments was: *n* = 3 for Control + PBS group, *n* = 5 for Control + LPS group, *n* = 4 for SMNΔ7 + PBS group, and *n* = 5 for SMNΔ7 + LPS group. For each specimen of each experimental animal, two tissue sections were used. Analysis power was calculated with GPower software (V. 3.1). Analyses blinded to the disease status were performed for all experimental procedures to ensure high data quality. The only tests which could not be performed in a blinded fashion were the neuromuscular evaluation clinical tests, as the pup size was indicative of the disease status. For behavioral analysis and the assessment of motor deficits, each procedure was quantified based on a specific rating scale. All data are expressed as means ± standard error of mean (SEM) unless otherwise stated. Statistical analysis was performed using PRISM (GraphPad, V 8.0.2) software. Both parametric and non-parametric tests were applied depending on the data input (one-way ANOVA or Kruskal–Wallis test, respectively) with the related Dunnett’s and Dunn’s multiple comparisons tests, respectively, and the level of statistical significance (alpha) was set at *p* < 0.05 (* *p*  <  0.05, ** *p*  <  0.01, *** *p*  <  0.001, **** *p* < 0.0001).

## 3. Results

### 3.1. Systemic E. coli LPS Administration Does Not Affect Clinical Manifestations of Experimental SMA

The experimental setting is presented in Figure 1A for clinical signs of disease progression and evaluation of disease symptoms. SMNΔ7 groups retained their weight within the expected disease-related spectrum irrespective of LPS administration, and weight gain was comparable to both control groups (Figure 1B). Notably, a significant difference with respect to weight (in grams) was observed between the PBS-injected SMNΔ7 mice and the PBS-injected control mice, indicative of an impact of the genotype on disease severity (2.67 ± 0.1 vs. 4.9 ± 0.57, * *p* = 0.049). For LPS-injected groups, the results were comparable to these of the PBS-injected groups, both for SMNΔ7 and controls (2.65 ± 0.09 vs. 4.69 ± 0.55, *p* = 0.06). Administration of LPS did not affect the performance of the SMNΔ7 or the control mice in the tail suspension test (Figure 1C,D). The mean scores of the SMNΔ7 LPS- and PBS-treated mice on this test were 1.61 ± 0.51 vs. 1.78 ± 0.49 (*p* > 0.99), respectively, while the scores for the LPS- and PBS- treated control mice were 3.94 ± 0.03 vs. 3.92 ± 0.05 (*p* > 0.99), respectively. Similarly to the data obtained from the suspension test (Figure 1E,F) no statistically significant difference was found between the control group that received LPS and those that received only PBS (mean score 4.0 ± 0.0 vs. 4.0 ± 0.0, *p* > 0.99), nor for the SMNΔ7 (mean score 1.94 ± 0.46 vs. 2.0 ± 0.4, *p* > 0.99).

### 3.2. LPS Systemic Administration Promotes Reduction in SMN Protein Levels in the CNS and Peripheral Organs

We further investigated SMN expression levels in CNS and peripheral tissues by means of WB to study whether the LPS-injected neonates differed from PBS-injected neonates in terms of SMN expression levels (Figure 2A). Regarding CNS tissues, the brain and spinal cord were investigated. For SMNΔ7 brains (Figure 2B), LPS-injected mice exhibited reduced SMN relative expression compared to the PBS-injected mice (0.42 ± 0.01 vs. 0.61 ± 0.03, respectively, **** *p* < 0.0001). Interestingly, a similar effect was evident for control mice (0.92 ± 0.007 vs. 1.0 ± 0.0, LPS-injected vs. PBS-injected, respectively, * *p* = 0.03), though at a lesser degree of significance. The results were similar for SMNΔ7 spinal cords (Figure 2C), where LPS-injected mice showed reduced SMN expression levels compared to the PBS-injected mice (0.23 ± 0.01 vs. 0.34 ± 0.01, respectively, **** *p* < 0.0001). A similar effect was evident for control mice (0.93 ± 0.01 vs. 1.0 ± 0.0, LPS-injected vs. PBS-injected, respectively, * *p* = 0.02). For the SMNΔ7 gastrocnemius muscle, the results are presented in Figure 2D, with LPS-injected mice demonstrating reduced SMN expression compared to the PBS-injected mice (0.31 ± 0.01 vs. 0.52 ± 0.02, respectively, **** *p* < 0.0001. The same trend was evident for control mice (0.9 ± 0.01 vs. 1.0 ± 0.0, for LPS-injected vs. PBS-injected, respectively, ** *p* = 0.003). For SMNΔ7 triceps brachii muscles (Figure 2E), LPS-injected mice exhibited reduced SMN expression compared to the PBS-injected mice (0.34 ± 0.02 vs. 0.51 ± 0.02, respectively, **** *p* < 0.0001). A similar effect was observed for control mice (0.89 ± 0.01 vs. 1.0 ± 0.0, respectively, **** *p* < 0.0001), though at a lesser degree of significance. For the SMNΔ7 gut (Figure 2F), LPS-injected mice exhibited reduced SMN relative expression compared to the PBS-injected mice (0.65 ± 0.02 vs. 0.91 ± 0.02, respectively, ** *p* = 0.008). Interestingly, a similar effect was evident for control mice (0.92 ± 0.01 vs. 1.0 ± 0.0, respectively, ** *p* = 0.06). For the spleen, there was no effect of LPS administration on the SMN relative expression among SMNΔ7 groups (0.9 ± 0.009 vs. 0.90 ± 0.006, *p* > 0.99) (Figure 2G). The results for control groups were comparable as well (1.0 ± 0.0 vs. 0.95 ± 0.007, *p* = 0.058, without vs. with LPS).

Further, in order to examine whether the CNS or periphery was more susceptible to LPS challenge, SMN ratio was compared between LPS- and PBS-injected mice, separately for the controls and SMNΔ7 mice for each tissue (Figure 2H). With respect to the control groups, the SMN ratio of CNS tissues for LPS- to PBS-injected mice was compared to peripheral organs. The results for the gastrocnemius muscle (0.95 ± 0.011 vs. 0.88 ± 0.014, for brain vs. gastrocnemius, *** *p* < 0.001), the triceps brachii muscle (0.95 ± 0.011 vs. 0.88 ± 0.010, for brain vs. triceps brachii, *** *p* < 0.001), and the gut (0.95 ± 0.011 vs. 0.84 ± 0.020, for brain vs. gut, *** *p* < 0.001) indicate that the periphery was, in fact, more susceptible to LPS administration in control mice. However, there was no difference found between the spleen and the brain (0.95 ± 0.011 vs. 0.96 ± 0.005, *p* = 0.6461). Regarding the comparison of the SMN ratio for LPS- to PBS-injected control mice in spinal cords with peripheral organs, differences were observed for the gastrocnemius muscle (0.93 ± 0.010 vs. 0.88 ± 0.014, for spinal cord vs. gastrocnemius, *** *p* < 0.001) and the triceps brachii muscle as well (0.93 ± 0.010 vs. 0.88 ± 0.010, for spinal cord vs. triceps brachii, *** *p* < 0.001). A similar effect was also noted for the gut (0.93 ± 0.010 vs. 0.84 ± 0.020, for spinal cord vs. gut, *** *p* < 0.001), but it did not extend to the spleen (0.93 ± 0.010 vs. 0.96 ± 0.005, for spinal cord vs. spleen, *p* = 0.0586). Concerning SMNΔ7 groups, results are presented as the SMN ratio for LPS- to PBS-injected mice as well. Regarding the SMN ratio for the brain in the SMNΔ7 groups, comparison to the gastrocnemius muscle (0.78 + 0.011 vs. 0.687 + 0.014, for brain vs. gastrocnemius muscle, *** *p* < 0.001), triceps brachii (0.78 + 0.011 vs. 0.658 + 0.010, for brain vs. triceps brachii, *** *p* < 0.001), gut (0.78 + 0.011 vs. 0.713 + 0.020, for brain vs. gut, *** *p* < 0.001), and spleen (0.78 + 0.011 vs. 0.956 + 0.005, for brain vs. spleen, *** *p* < 0.001) revealed differences between the susceptibility of CNS and periphery to LPS-induced SMN reduction. Regarding the SMN ratio for the spinal cord in SMNΔ7 groups, a similar effect was detected for both the gastrocnemius muscle (0.76 + 0.010 vs. 0.687 + 0.014, for spinal cord vs. gastrocnemius, *** *p* < 0.001) and the triceps brachii muscle (0.76 + 0.010 vs. 0.658 + 0.010, for spinal cord vs. triceps brachii, *** *p* < 0.001). Furthermore, a comparison of the SMN ratio for SMNΔ7 groups concerning spinal cord and gut results showed a difference between the susceptibility of the CNS and periphery to LPS challenge (0.76 + 0.010 vs. 0.713 + 0.020, for spinal cord vs. gut, *** *p* < 0.001). A similar effect was evident for the spleen as well (0.76 + 0.010 vs. 0.956 + 0.005, for spinal cord vs. spleen, *** *p* < 0.001). Overall, these data suggest an effect of systemic LPS administration leading to a reduction in SMN protein expression levels in the CNS, the skeletal muscle, and the gut, tissues with high energy consumption and increased demand in terms of structural prerequisites.

### 3.3. LPS Administration Aggravates Axonal Loss and Leads to Reduced Immunoreactivity of SYP-Positive Synapses in the CNS of SMNΔ7 Mice

To further examine the number of neurons, we performed Nissl staining on spinal cords (Figure 3A–E), and no statistically significant differences were found following LPS administration, either in the control group (73.2 ± 6.93 vs. 77.1 ± 2.98, *p* > 0.99, with vs. without LPS administration, respectively) or in the SMNΔ7 mice (33.2 ± 4.06 vs. 33.7 ± 2.65, *p* > 0.99, with vs. without LPS administration, respectively). We performed Nissl staining on the spinal cords to assess the neuron number and determined that LPS administration did not result in a significant difference in motor neuron numbers in either the SMNΔ7 or control groups (Figure 3A–E); however, the control spinal cords had more than twice as many motor neurons per unit area. SMNΔ7 mice treated with LPS or PBS had 33.2 ± 4.06 and 33.7 ± 2.65 motor neurons/mm^2^, respectively (*p* > 0.99), while the LPS treated control mice had 73.2 ± 6.93 and those treated with PBS had 77.1 ± 2.98 motor neurons/mm^2^ (*p* > 0.99). Similarly, for the brainstem sections (Figure 3F–J), there was no statistically significant difference following LPS administration, either in the control group (MNs/mm^2^: 31.7 ± 2.35 vs. 33.0 ± 2.25, *p* > 0.99, with vs. without LPS administration, respectively) or in the SMNΔ7 group (MNs/mm^2^: 20.6 ± 3.29 vs. 25.4 ± 4.05, *p* > 0.99 with vs. without LPS administration, respectively). In contrast, a significant effect of LPS was found on axonal damage in both control and SMNΔ7 mice. More specifically, axonal loss was assessed by Bielschowsky staining in the spinal cord (Figure 3K–O), and mice that received LPS exhibited reduced axonal density in both the control (mean score: 1.17 ± 0.24 vs. 0.16 ± 0.11, * *p* = 0.04, with vs. without LPS administration, respectively) and the SMNΔ7 groups (mean score: 2.17 ± 0.2 vs. 1.0 ± 0.0, * *p* = 0.03, with and without LPS administration, respectively). Similarly, for the brainstem (Figure 3P–T), mice that received LPS exhibited reduced axonal density in both the control (mean score: 0.5 ± 0.23 vs. 0.16 ± 0.11, *p* > 0.99, with vs. without LPS administration, respectively) and the SMNΔ7 groups (Mean score: 1.42 ± 0.26 vs. 0.41 ± 0.19, * *p* = 0.03, with and without LPS administration, respectively). Our results indicate that LPS systemic administration promoted increased axonal loss not only in mice with SMA, but also in the control mice.

Alterations in synaptic density were investigated with immunofluorescence reactivity for synaptophysin (SYP). In the spinal cords, LPS administration promoted reduction of SYP density in the control group (Figure 4B) compared with the PBS-injected mice (Figure 4A) (integrated density: 1.9 × 10^6^ ± 0.14 × 10^5^ vs. 2.2 × 10^6^ ± 0.16 × 10^5^, * *p* = 0.04, with vs. without LPS). Similarly, in the spinal cords of SMNΔ7 mice, LPS administration had the same effect, where LPS-injected mice exhibited decreased SYP density (Figure 4D) compared to PBS-injected mice (Figure 4C) (integrated density: 1.5 × 10^6^ ± 0.13 × 10^5^ vs. 1.8 × 10^6^ ± 0.17 × 10^5^, * *p* = 0.04), with vs. without LPS, respectively). The results for spinal cords are presented in Figure 4E. Upon SYP investigation in the brainstem (Figure 4F–I), no statistically significant differences were found following LPS administration, either in the control group (integrated density: 2.1 × 10^6^ ± 0.24 × 10^5^ vs. 2.2 × 10^6^ ± 0.006 × 10^5^, *p* = 0.26, with vs. without LPS, respectively) or in the SMNΔ7 mice. SYP levels for SMNΔ7 mice were found to be elevated in the brainstems of PBS-injected mice in comparison to the LPS-injected group (integrated density: 1.7 × 10^6^ ± 0.29 × 10^5^ vs. 2.0 × 10^6^ ± 0.17 × 10^5^, * *p* = 0.04, with vs. without LPS). The results for SYP reactivity in the brainstem are demonstrated in Figure 4J.

### 3.4. LPS Administration Promotes a Reactive Phenotype in the Microglia and Astrocytes of SMNΔ7 Mice

Iba-1 was used to quantify microglia in late-symptomatic SMNΔ7 mice and control mice, both with and without LPS administration. Both the microglial cell count and morphology were studied. Microglial cells’ quantification results revealed that LPS administration promoted an increase in the Iba-1^+^ microglial cell population in both CNS regions examined. In the spinal cords, an increase in microglial cell count was evident in response to LPS administration in the control group (80.2 ± 2.72 vs. 60.1 ± 4.17, * *p* = 0.025, with vs. without LPS administration, respectively) and in the SMNΔ7 group (103.0 ± 4.95 vs. 83.1 ± 4.38, * *p* = 0.046, with vs. without LPS administration, respectively) (Figure 5A–E). In the brainstems, almost the same trend was detected, as the PBS-injected control group had a lower number of Iba-1^+^ microglial cells compared with LPS-injected control mice (92.9 ± 3.7 vs. 70.1 ± 4.6, * *p* = 0.012, with vs. without LPS administration), as shown in Figure 5F,G. LPS-injected mice from the SMNΔ7 group also had elevated microglial populations in comparison to the PBS-injected SMNΔ7 group (124 ± 3.43 vs. 111.0 ± 3.67, * *p* = 0.031, with vs. without LPS administration) (Figure 5H–J). Regarding the morphological alterations observed in the spinal cords, our results show that, in control mice, LPS administration promoted alterations in microglial cell morphology, namely, increased cell body area and a reduction in microglial branches, evidenced as an increase in the ramification index (2.05 ± 0.07 vs. 1.23 ± 0.09, ** *p* = 0.007, LPS-injected vs. PBS-injected control mice) (Figure 5U). Furthermore, following LPS administration, the same effect was observed in SMNΔ7 mice, although to a lesser extent. In the spinal cords, an increase in the ramification index was detected in LPS-injected SMNΔ7 mice compared to PBS-injected SMNΔ7 mice (2.66 ± 0.05 vs. 2.16 ± 0.06, * *p* = 0.03, LPS-injected vs. PBS-injected SMNΔ7 mice). In the brainstems, morphological analysis revealed a similar effect after LPS administration in control mice (Figure 5V) with an increase in the ramification index, evidenced as an enlargement of the cell body area and limited branches of Iba-1^+^ microglial cells (1.86 ± 0.03 vs. 1.48 ± 0.05, * *p* = 0.01, LPS-injected vs. PBS-injected control mice). Similar results were detected in SMNΔ7 mice as well, where LPS administration promoted cell body enlargement and elimination of microglial branches, which led to an increased ramification index (2.21 ± 0.06 vs. 1.86 ± 0.05, * *p* = 0.01, LPS-injected vs. PBS-injected SMNΔ7 mice). In conclusion, both quantitative and morphological alterations were noted in the microglia population after LPS administration.

Regarding the astrocyte count in CNS following LPS or PBS administration, the GFAP marker was used to evaluate astrocyte populations. Increased numbers of GFAP^+^ astrocytes were detected in the LPS-injected control group compared to PBS-injected mice (212.0 ± 2.94 vs. 183.0 ± 2.32, ** *p* = 0.004, with and without LPS, respectively), as shown in Figure 5K,L for the spinal cords. The results for the SMNΔ7 group detected a mild increase in LPS-injected mice compared to PBS-injected pups (270.0 ± 3.70 vs. 240.0 ± 3.77, * *p* = 0.03, with vs. without LPS). In the brainstems, increased numbers of GFAP^+^ astrocytes in the LPS-injected control group compared to the PBS-injected control mice (134.0 ± 2.6 vs. 125.0 ± 3.04, * *p* = 0.03 with vs. without LPS) were detected, as shown in Figure 5P,Q. Furthermore, increased numbers of GFAP^+^ astrocytes were also evident in LPS-injected compared to PBS-injected mice (144.0 ± 2.38 vs. 136.0 ± 3.1, * *p* = 0.01, with vs. without LPS). The results are presented in Figure 5R–T.

### 3.5. LPS Administration Promotes Predominance of Innate Immunity Cellular Populations with Neurotoxic Phenotype in the CNS of SMNΔ7 Mice

To further characterize the activated phenotype of innate immunity cellular populations, we performed double immunofluorescence staining. In order to detect pro-inflammatory microglia, Iba-1^+^iNOS^+^ cells were counted. LPS-injected control mice exhibited increased numbers of pro-inflammatory Iba-1^+^iNOS^+^ microglia in the spinal cord compared to PBS-injected control mice (47.3 ± 4.42 vs. 22.5 ± 1.71, *p* < 0.0001) (Figure 6A,B). Similarly, for the SMNΔ7 mice, LPS administration was associated with increased numbers of Iba-1^+^iNOS^+^ cells (69.4 ± 5.78 vs. 35.4 ± 2.0, *p* = 0.003) compared to PBS-injected mice (Figure 6C–E). In the brainstem, PBS-injected controls expressed decreased numbers of Iba-1^+^iNOS^+^ reactive microglial cells compared to the LPS-injected group (31.9 ± 1.3 vs. 61.6 ± 2.84, **** *p* < 0.0001, respectively). LPS administration promoted enhanced expression of Iba-1^+^iNOS^+^ microglial population in the SMNΔ7 group as well in comparison to the PBS-injected SMNΔ7 group (78.8 ± 1.64 vs. 45.9 ± 3.61, **** *p* < 0.0001). Results are presented in Figure 6F–J. Investigation of the anti-inflammatory microglial population was conducted with the Iba-1^+^CD206^+^ cell count in CNS. Results for the spinal cords are presented in Figure 6K–O, where the data demonstrate comparable Iba-1^+^CD206^+^ cell counts after LPS or PBS administration in the control group (47.3 ± 3.51 vs. 46.5 ± 1.83, respectively, *p* > 0.99). Although, in the SMNΔ7 group, LPS administration reduced the Iba-1^+^CD206^+^ microglial population, the noted difference did not reach statistical significance (137 ± 5.78 vs. 94.6 ± 2.41, *p* = 0.29, without vs. with LPS). Results for the brainstems are presented in Figure 6P–T. Sub-populations of anti-inflammatory microglial Iba-1^+^CD206^+^ cells were found to be elevated in PBS-injected control mice compared to the LPS-injected group (115 ± 6.28 vs. 101 ± 3.86, *p* = 0.27). Similarly, the PBS-injected SMNΔ7 group showed increased numbers of Iba-1^+^CD206^+^ cells in comparison to LPS-injected SMNΔ7 mice (121.0 ± 3.48 vs. 108 ± 3.87, *p* = 0.27). 

Regarding the astrocyte contribution in the CNS of the SMNΔ7 model, astrocytes with pro-inflammatory phenotypes were detected with GFAP and complement C3 markers. The results for the spinal cords showed an increased presence of pro-inflammatory astrocyte sub-populations in the LPS-injected group compared to the PBS-injected mice (10.3 ± 0.14 and 9.08 ± 0.09 for LPS-injected and PBS-injected group, ** *p* = 0.001, respectively) (Figure 7A,B). Similarly, the LPS-injected SMNΔ7 group showed an increased presence of GFAP^+^C3^+^ astrocytes compared to the PBS-injected SMNΔ7 mice (Figure 7C–E) (13.0 ± 0.25 vs. 11.6 ± 0.24, * *p* = 0.049, prior-to vs. after LPS). Regarding the brainstems, results are presented in Figure 7F–J, and were comparable between the LPS-injected and PBS-injected control groups (4.85 ± 0.2 vs. 4.25 ± 0.09, *p* = 0.056, respectively). For the LPS-injected SMNΔ7 group, the results for GFAP^+^C3^+^ astrocytes were similar to those in the PBS-injected SMNΔ7 mice (5.27 ± 0.21 vs. 5.06 ± 0.16, *p* > 0.99, respectively). Astrocytes with anti-inflammatory phenotypes were also counted with the GFAP and S100A10 markers. For the spinal cords, LPS administration promoted a reduction in the beneficial astrocyte population, as shown in Figure 7K–O. The results for GFAP^+^S100A10^+^ astrocytes revealed reduced expression in the LPS-injected control group compared to the PBS-injected group (214.0 ± 3.03 vs. 233.0 ± 5.21, * *p* = 0.039, respectively). A reduction was also detected in the LPS-injected SMNΔ7 group compared to the PBS-injected SMNΔ7 mice (213 ± 2.71 vs. 237 ± 4.07, ** *p* = 0.004). In the brainstem region (Figure 7P–T), no significant difference was found after LPS administration. The results for the LPS-injected control group were comparable to the PBS-injected control mice (72.1 ± 4.08 vs. 73.2 ± 3.44, *p* > 0.99). For the LPS-injected SMNΔ7 group, the results for GFAP^+^S100A10^+^ astrocytes were similar to those in the PBS-injected SMNΔ7 mice (61.9 ± 3.25 vs. 65.5 ± 3.11, *p* > 0.99). Overall, these data show that systemic administration of LPS induces reactivity of the glial cells in experimental SMA, thus further supporting the link between systemic inflammation and neuroinflammation driving neurodegeneration.

## 4. Discussion

SMA is an interesting intersection between the brain, gut, and skeletal muscles; however, the mechanisms underlying their communication remain under investigation. Environmental factors have been implicated in the development of diseases, either primary neurodegenerative, such as Alzheimer’s disease (AD), SMA, and ALS [34,35] or immune-mediated and neurodegenerative, such as multiple sclerosis [36,37]. The gut has been recognized as an organ that, under microbial stimuli, may regulate immune cell populations, with implications for CNS immune-mediated diseases [38]. Recently, the neuro-inflammatory component in the pathology of otherwise primary neurodegenerative diseases has emerged as the focus of extensive research [39,40]. Cells of the innate immune system of the CNS, namely, microglia and astrocytes, are primary mediators of this neuro-inflammatory component, and their exact contribution to disease remains elusive. Moreover, these cellular populations are highly subject to environmental stimuli, thus constituting a potentially modifiable component of the disease pathology. In order to elucidate the possible involvement of a microbial antigen in SMA and its possible effect on the neuro-inflammatory component, we applied an experimental approach of systemic bacterial LPS administration in SMNΔ7 mice.

The SMNΔ7 mice used herein were selected as a widely used, well-characterized, and universally accepted mouse model of SMA. However, it is a model of increased severity (simulating type II human SMA) with an especially short survival time. Therefore, its phenotype is subject to severe neurodegenerative pathology early in the disease course. This is an inherent limitation of our study, and for this reason, further investigation of the LPS systemic administration effect in a model of milder SMA phenotype is warranted. Moreover, as previously described for the battery of motor tests used in SMA experimental models and in patients, ceiling and floor effects are very common; therefore, it is likely that most of the tests applied may be unable to discriminate mild alterations in motor behavior [41]. An additional limitation of our study is the small sample size per experimental group, a common limitation of studies conducted with transgenic mice exhibiting harmful phenotypes, with a limited number of viable offspring produced. In the present study, triple transgenic mice of the SMNΔ7 model produced only a limited number of mice with SMA from each birth, and they were not always able to survive until the late-symptomatic stage and died soon after birth due to SMN deficiency. In order to compensate for this limitation, we attempted to combine data from two independent experiments. 

In the present study, 5 μg E. coli LPS was administered via intra-peritoneal injection to neonates on postnatal day 5 (P5). Although 5 μg is a sub-lethal dose, it is a very high dose of LPS for mice on postnatal day 5, in particular for these SMNΔ7 mice. Limited experimental approaches utilize LPS challenge in SMA models. More specifically, currently available data regarding LPS administration in SMA neonates are presented in [42]. According to data retrieved from this experimental approach, postnatal neonates at day 1 (PND1) which received a dose of 6 μg/g survived for almost 50 h, as reported for the SMA group, and for the control mice, the survival rate was almost 78 h following injection. Naturally, a limited lifespan makes sense after taking into consideration that the Taiwanese model in use has a maximum survival life of eight to ten days. As far as we can tell, the reason for these discrepancies between this experiment and our approach could be related to the different day of administration. Therefore, for mice Taiwanese model, LPS administration was performed at PND1, which is considered a pre-symptomatic stage, and seems to affect the manifestation of disease more severely. Instead, in our approach, LPS challenge was induced at PND5 in the SMNdelta7 model, which is considered an early-onset stage for this model and did not seem to affect the phenotypic manifestation of disease in the context of neuromuscular performance. In addition, genuine differences between these models may have also contributed to the observed differences. For example, the Taiwanese model exhibited a more severe phenotype with limited survival compared to SMNdelta 7. Notably, in the present study, a smaller sample size per group was used compared to the aforementioned study. Furthermore, the results of [43], which used neonates of a different strain, also demonstrated that the newborns survived for up to nine days after LPS administration. More precisely, neonates from PND4-PND6 that were repeatedly injected with 6 μg/g of intraperitoneal LPS survived for up to 9 days after the injection, at which point they were euthanized [43]. Collectively, it is reasonable to note that the impact of LPS challenge may be closely related to the day of administration, the strain used, and the dosage utilized in each experimental method.

In our study, LPS administration did not induce a prominent effect in experimental SMA in terms of clinical outcomes. However, LPS systemic administration induced a potent diminishing effect on SMN expression in SMNΔ7 mice, affecting the skeletal muscles and the CNS, as well as the gut, whereas a similar effect was not evident in the spleen. Organs are known to grow proportionally with the body, and by the model’s nature, the SMA mice presented smaller body sizes compared to the control mice. Interestingly, although SMN protein levels differed among experimental groups for the skeletal muscle and the gut, we did not observe a similar significant difference across experimental groups in the spleen. Notably, the sizes of the spleens in SMNΔ7 mice were significantly smaller than those in control mice. Also, SMNΔ7 mice displayed alterations in lymphocyte concentration and macrophage population [44]. However, overall, the spleen has been reported to exhibit higher levels of the SMN protein than other organs [45], and this difference may account for the lack of observed differences across experimental groups. Moreover, the short lifespan of the SMNΔ7 model may not have permitted the development of the full disease phenotype in all organs, especially organs with increased SMN protein abundance, such as the spleen. Notably, a reduction in spleen size was reported in experimental SMA, with the milder Smn^2B/−^ model exhibiting the most drastic splenic atrophy, in contrast to the more severe mouse model [45]. With respect to the effect of LPS administration on experimental SMA, our results are in agreement with previous reports in a severe SMA model, where LPS-induced systemic inflammation promoted a reduction in SMN levels in CNS and periphery [42]. Importantly, in our study, the effect of LPS administration on SMN expression was prominent in experimental SMA, contrary to control mice, where a similar effect was evident to a lesser extent. This observation underlines the fact that, in a mouse model of neurodegeneration induced by a genetically determined reduction in the SMN expression, systemic microbial stimuli may act as factors that further enhance model pathology. 

LPS-mediated modifications of alternative splicing and gene transcription have previously been shown to lead to a decrease in SMN protein levels in the CNS and peripheral tissues [42,46,47,48,49,50]. A different mechanism linked to LPS-induced protein damage may be the abnormal homeostasis of the ubiquitin–proteasome system [51,52,53]. In particular, exposure to LPS amplifies the activation of the ubiquitin–proteasome system, which in turn causes subsequent muscle damage and protein degradation in the skeletal muscles [54,55]. The relative balance between protein synthesis and protein breakdown determines skeletal muscle atrophy [56]; thus, during a muscle atrophy state, protein degradation exceeds protein synthesis. Moreover, degradation of myofibrillar proteins has also been reported after exposure to LPS stimulus [57]. Excess atrophy in C2C12 myotubes has been linked to LPS-induced TLR4 receptor activation. It is speculated that the detrimental effects of LPS are mediated via up-regulation of autophagosome formation and ubiquitin ligase expression, which cause excessive proteolysis, particularly in the skeletal muscles [9,58]. More specifically, the SMN protein, derived both from full-length and SMNΔ7 transcripts, is known to be mono-ubiquitinylated [59]. SMN is strictly regulated and interacts directly with its ligase in order to establish cellular localization and to preserve appropriate function [59]. However, after polyubiquitinylation, as occurs in the presence of LPS, the SMNΔ7 transcripts, which are inherently unstable, become more vulnerable to proteolysis and are rapidly degraded [60]. Intriguingly, mutations of the ubiquitin-like modifier activating enzyme 1 (UBA1) gene, which disrupts the normal ubiquitinylation pattern, are associated with a rare X-linked SMA form and are implicated in other neurodegenerative diseases as well, such as Huntington’s chorea. (reviewed in [61]). The exact mechanism of LPS-induced proteolysis in association with the muscle SMN expression and the related implication for SMA remains to be elucidated. 

Microbial antigens may also affect skeletal muscle directly via immune-mediated mechanisms [62]. More specifically, experimental evidence indicates that LPS induces increased chemokine production in skeletal muscle, such as monocyte chemoattractant protein 1 expression (MCP-1) [63], mediated by the nuclear factor kappa B (NF-κB) pathway. NF-κB activation appears to be almost twenty times stronger for TLR4 ligands, such as LPS, compared to other TLR ligands, such as TLR2. Excessive activation of the NF-κB pathway has been correlated with a loss of skeletal muscle mass in mice [5]. Data from an experimental SMA model demonstrated that LPS administration, in addition to inducing a strong inhibition of exon 7 inclusion in the SMN2 gene transcript, resulting in a decrease of SMN protein levels in both peripheral and CNS tissues, also promoted the rapid development of systemic inflammation [42]. This systemic inflammation was due to endotoxin leakage from the gut as a result of compromised intestinal barrier integrity in experimental mice [42]. Prominent systemic inflammation caused by increased circulating endotoxins has also been observed in patients with skeletal muscle atrophy, and this implies a critical role of activation of inflammatory pathways that lead to the establishment of muscle atrophy through myokine production [64]. Moreover, LPS-induced TLR signaling mediated by the NF-κB pathway has been implicated in mitochondrial function and homeostasis, as it has been shown to indirectly promote the production of oxidative radicals in mitochondria [65].

With respect to the effect of systemic LPS administration in the CNS of mice with experimental SMA, in our study, LPS administration did not promote MN loss in mice with experimental SMA or control mice, as assessed by Nissl staining. However, based on our results, LPS administration deteriorated the expression of presynaptic protein synaptophysin in the CNS of mice with experimental SMA. Afferent synapse loss and the reduced proprioceptive reflexes of SMA MNs are still under investigation [66]. Presynaptic terminal deficits may be associated with excitability fluctuations in MN synaptic transmission in SMA, as demonstrated by Sun and Harrington, who showed that inhibitory synaptic input was diminished in MNs, but elevated in interneurons of the SMN∆7 model [67]. Also, the reduced sensory–motor excitatory synaptic transmission could be implicated in the subsequent reduction found in motor neuron firing and the decrease in the expression of the potassium channel Kv2.1 on the surfaces of MNs [68]. Moreover, reports of hyperexcitability in MNs and sensory–motor neurons are supported by data retrieved from patient studies and SMA models as well [66,67,69,70]. A growing body of evidence is establishing the emerging role of spinal interneurons as gatekeepers to neuroplasticity after injury or disease [71]. Spinal interneurons constitute a diverse population that form inhibitory microcircuits and display neuronal input specificity tailored to individual limb muscles. With respect to these properties, interneurons are promising targets which are amenable to intervention in the context of diseases such as SMA, where specific neurons are found to be selectively vulnerable. A recent study revealed the beneficial potential of large cholinergic synapse modification for motoneurons (V0c interneurons) in ALS [72,73]. Furthermore, data from the SODG93A ALS mouse model reported that the aberrant synaptic connectivity of V1 subtypes increased the susceptibility of motoneurons to degeneration during disease progression [74].

The main mechanisms proposed for aberrant synaptic transmission are deficits in axonal transport and microglia-mediated dysfunction. Experimental data support the existence of dynamic changes in axonal synaptic vesicle-associated proteins and alterations in retrograde transport that lead to changes in pre-synaptic markers like synaptophysin after LPS administration in Parkinson’s Disease, leading to the degeneration of dopaninergic neurons [75]. This could be explained by the long distance between cell bodies and distal axons in these neurons, as sensory and motor neurons appear to be particularly susceptible to disruptions in axonal transport [76]. The second mechanism of LPS-induced presynaptic dysfunction involves activated microglia. Activated microglia, after the inflammatory insult of LPS, can act directly within brain tissue. LPS-induced neurodegeneration and synaptic damage have been established previously in vitro [77], and this effect has been shown to be microglial-dependent and associated with microglia-derived interleukin-1beta (IL1β) or tumor necrosis factor alpha (TNF-a) levels [77,78]. Further insights into the effect of innate immunity cells in neurodegenerative diseases, including SMA, have also linked glial cells [45,79,80,81] with neuroinflammation leading to neurodegeneration [82,83]. Existing evidence underlines the implication of astrocytes and microglia in SMA pathogenesis [80], and these populations are susceptible to the effect of systemic microbial stimuli [84]. Notably, in vivo evidence supports the contribution of LPS-induced microglial activation to other neurodegenerative diseases, such as ALS [85] and an experimental LPS-induced AD-like model of cognitive decline [86]. In the latter model, LPS-induced synapse damage was linked to microglia activation, mediated through the classical complement pathway. Thus, microglia activation appears to be a converging point between neurodegeneration and neuroinflammation, as is evident in several CNS pathologies. 

In our study, morphological changes and increased numbers of astrocytes and microglia were detected following LPS administration in late-symptomatic mice with experimental SMA. Morphological alterations included cell body enlargement and reductions in branch number and length, as was observed in both aforementioned populations. These changes are related to activated states of microglia and astrocytes in response to peripheral stimuli and have been reported in several neurodegenerative diseases, orchestrating concurrent neuroinflammation [87,88]. Although microglia and astrocyte activation have been described in SMA, the exact role of these populations in the disease course and their phenotypic characterization is currently under investigation [89,90]. LPS systemic administration is known to rapidly promote pro-inflammatory microglia phenotypes in the CNS [91], and the bidirectional communication of these populations is a crucial mediator in terms of delaying or accelerating the establishment of neuroinflammation [92]. In SMA, aberrant communication of these populations may contribute to disease pathology [93]. In our study, LPS systemic administration was strongly correlated with an increase in CNS pro-inflammatory microglia and astrocytes (defined as Iba-1^+^iNOS^+^ and GFAP ^+^C3^+^ cells, respectively), and this effect, although present in both SMNΔ7 and control mice, was more prominent for mice with experimental SMA compared to control mice. These findings provide insight into the contributions of pro-inflammatory innate cell phenotypes to SMNΔ7 disease phenotype evolution when subjected to the effect of systemic microbial stimuli. Notably, in the SMNΔ7 model, the activation of innate immune cells with pro-inflammatory phenotypes may represent a bystander effect on the CNS in response to the primary neurodegeneration, as shown for several neurodegenerative diseases [83]. In this respect, the effect of LPS-induced alterations in the activation status and the CNS innate immune cell phenotype across the disease course warrants further investigation.

## 5. Conclusions

Our study provides thorough, mechanistic insight into the effect of systemic administration of microbial LPS on experimental SMA in terms of SMN expression, clinical disease outcomes, as well as the equilibrium between neurodegeneration and the activation of innate immunity in the CNS. Based on our results, we hereby provide evidence that systemic LPS administration in SMA further induces a profound decrease in SMN expression, worsens synapse pathology, and is associated with microglia and astrocyte alterations in terms of cell numbers, activation status, and reactive phenotype. Notably, whether the activation of innate immunity in the CNS in the context of experimental SMA further contributes to disease pathology remains to be elucidated. For this reason, a more thorough temporal evaluation of the pathology evolution in experimental SMA (neurodegeneration vs. neuroinflammation) at earlier time-points is necessary. The novelty of our study compared to previous studies utilizing the LPS challenge in SMA models is that it addresses the CNS impact more thoroughly and in a less severe (intermediate) type of disease. In the context of newly available treatments for SMA that effectively alter the disease phenotype towards milder clinical outcomes, the investigation of additional means of intervention, such as the modification of the CNS innate immune system, is of especially prominent importance.

## Figures and Tables

**Figure 1 cells-13-00785-f001:**
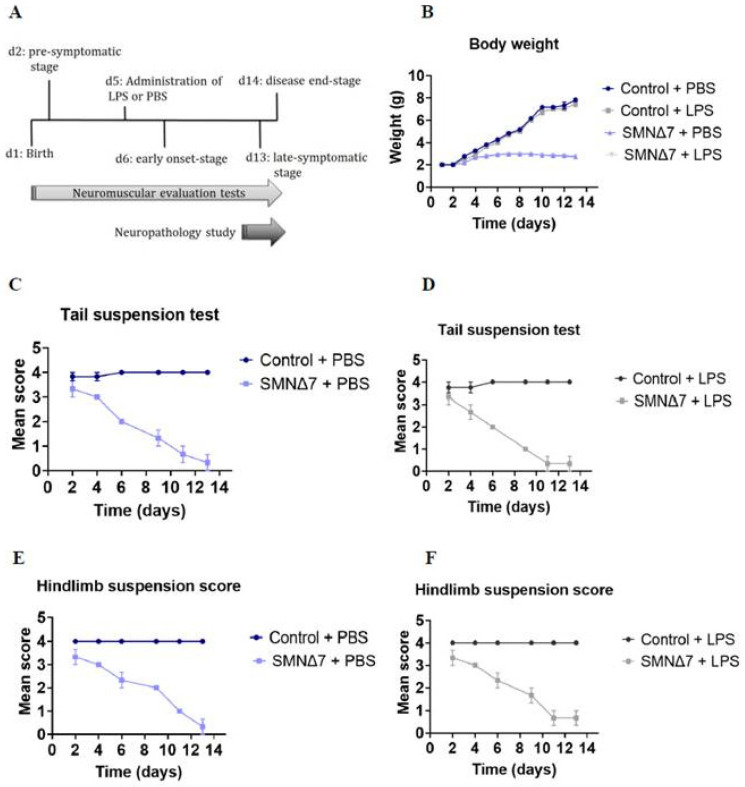
(**A**) Overview of experimental design and LPS intervention during time-course milestone events in SMNΔ7 mouse model and control mice. Evaluation of neuromuscular and disease-related symptoms was performed every other day during the experiment. Tissue harvesting was conducted in the late-symptomatic stage. (**B**) Mean body weight (in grams) across experimental groups during the first 13 postnatal days. (**C**) Mean score for tail suspension test across PBS-injected experimental groups during the first 13 postnatal days. (**D**) Mean score for tail suspension test across LPS-injected experimental groups during the first 13 postnatal days. (**E**) Mean score for hind limb suspension test across PBS-injected experimental groups during the first 13 postnatal days. (**F**) Mean score for hind limb suspension test across LPS-injected experimental groups during the first 13 postnatal days. For (**B**–**D**), control + PBS (dark blue), control + LPS (dark grey), SMNΔ7 + PBS (light blue) and SMNΔ7 + LPS (light gray). Error bars represent standard error of mean. Graphs show representative results of two independent experiments with *n* = 3 for Control + PBS group, *n* = 5 for Control + LPS group, *n* = 4 for SMNΔ7 + PBS group, and *n* = 5 for SMNΔ7 + LPS group.

**Figure 2 cells-13-00785-f002:**
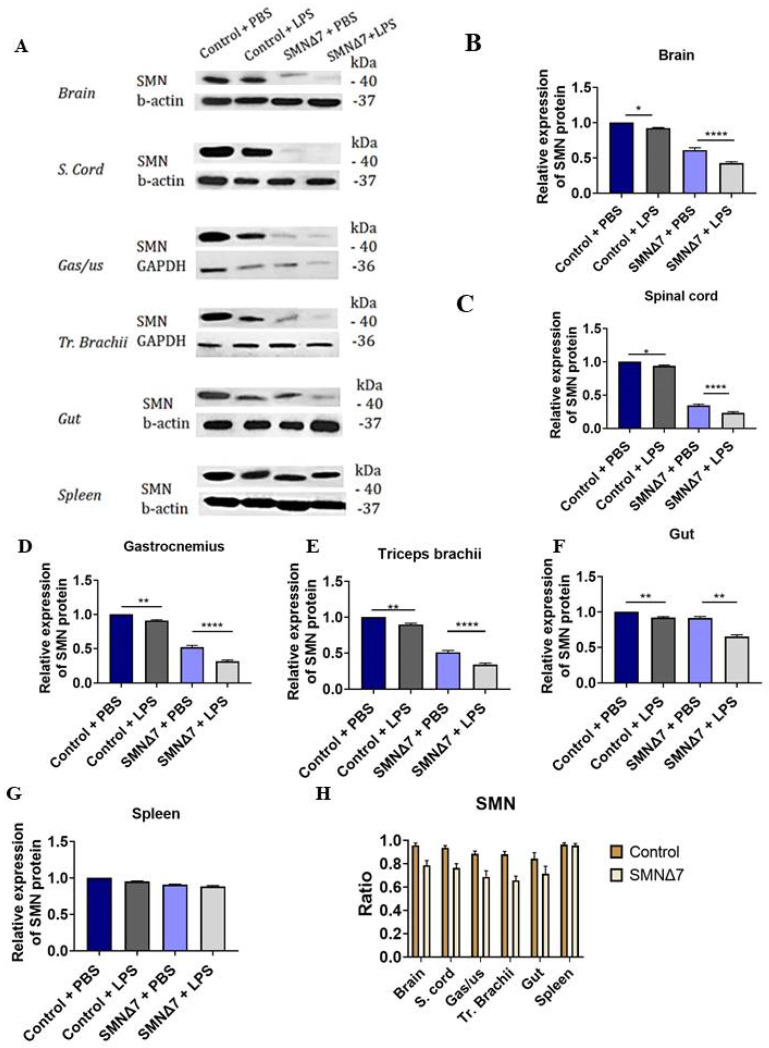
(**A**) Western blots for SMN, beta-actin, and GAPDH protein expression levels in CNS and peripheral tissues. Graphical representation of SMN protein expression levels across experimental groups: control + PBS (dark blue), control + LPS (dark grey), SMNΔ7 + PBS (light blue), and SMNΔ7 + LPS (light gray) in (**B**) brain, (**C**) spinal cord, (**D**) gastrocnemius muscles, (**E**) triceps brachii muscles, (**F**) gut, and (**G**) spleen. (**H**) Graphical representation of SMN ratio among different CNS and peripheral tissues. Bars represent mean ± standard error of the mean. Error bars represent standard error of the mean. SMN protein relative expression was assessed with beta-actin for CNS, gut, and spleen and GAPDH for muscle tissues. For the quantification of relative expression of SMN protein, SMN expression for control + PBS mice was calculated and set as 1. S. cord, spinal cord; gas/us, gastrocnemius; tr. branchii, triceps branchii. * *p* < 0.05; ** *p* < 0.005; **** *p* < 0.0001.

**Figure 3 cells-13-00785-f003:**
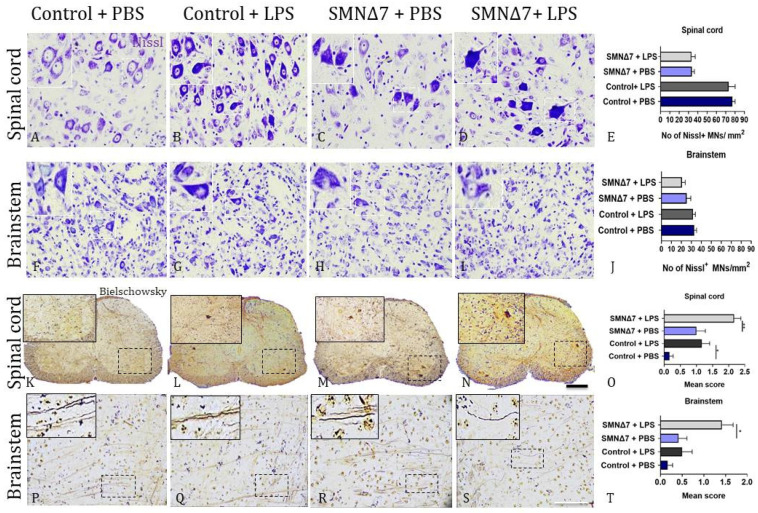
Representative histochemical results in CNS for MN count and estimation of neurodegenerative deposition with Nissl and Bielschowsky stain. (**A**–**D**) Representative results for Nissl+ MNs in spinal cord across experimental groups. (**E**) Graphical demonstration of Nissl + MN count in spinal cord. (**F**–**I**) Representative results for Nissl+ MNs in brainstem across experimental groups. (**J**) Graphical illustration of Nissl+ MN count in brainstem. (**K**–**N**) Representative results of axonal loss and degree of degeneration assessed by Bielschowsky staining in spinal cord across experimental groups. (**O**) Graphical illustration of degeneration defects in spinal cord region. Representative results of axonal loss and degree of degeneration assessed by Bielschowsky staining in brainstem for (**P**) PBS-injected control group, (**Q**) LPS-injected control group, (**R**) PBS-injected SMNΔ7 group and (**S**) LPS-injected SMNΔ7 group, respectively. (**T**) Graphical illustration of degeneration deficits in brainstem. Estimation of neurodegenerative deposition using Bielschowsky stain was conducted according to the following scores: 0: normal or even silver stain throughout the measured area; 1: small areas found to lack silver stain in at least one axon; 2: small but frequent areas that lack silver stain in at least one axon or apparent mild silver deposition; and 3: extensive loss of silver stain evident in at least one axon or severe silver deposition. Experimental groups: control + PBS (dark blue), control + LPS (dark grey), SMNΔ7 + PBS (light blue), and SMNΔ7 + LPS (light gray). Frames indicate inserts. Bars represent mean ± standard error of the mean. Error bars represent standard error of the mean. Graphs show representative results of two independent experiments with *n* = 3 for Control + PBS group, *n* = 5 for Control + LPS group, *n* = 4 for SMNΔ7 + PBS group, and *n* = 5 for SMNΔ7 + LPS group. * *p* < 0.05; ** *p* < 0.005; scales: 10×, 40×.

**Figure 4 cells-13-00785-f004:**
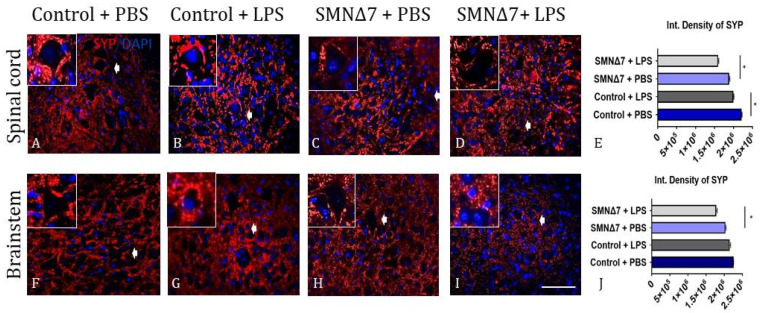
Representative results of immunofluorescence for estimation of pre-synaptic defects in CNS. (**A**–**D**) Results of synaptophysin (SYP) reactivity in spinal cord across experimental groups. (**E**) Graphical demonstration of SYP reactivity in spinal cord. (**F**–**I**) Results for SYP reactivity in brainstem across experimental groups. (**J**) Graphical demonstration of SYP reactivity in brainstem across experimental groups. Experimental groups: control + PBS (dark blue), control + LPS (dark grey), SMNΔ7 + PBS (light blue), and SMNΔ7 + LPS (light gray). Arrows indicate inserts. Bars represent mean ± standard error of mean. Error bars represent standard error of mean. Graphs show representative results of two independent experiments with *n* = 3 for Control + PBS group, *n* = 5 for Control + LPS group, *n* = 4 for SMNΔ7 + PBS group, and *n* = 5 for SMNΔ7 + LPS group. Int. Density of SYP: integrated density of synaptophysin; * *p* < 0.05; scale: 40×.

**Figure 5 cells-13-00785-f005:**
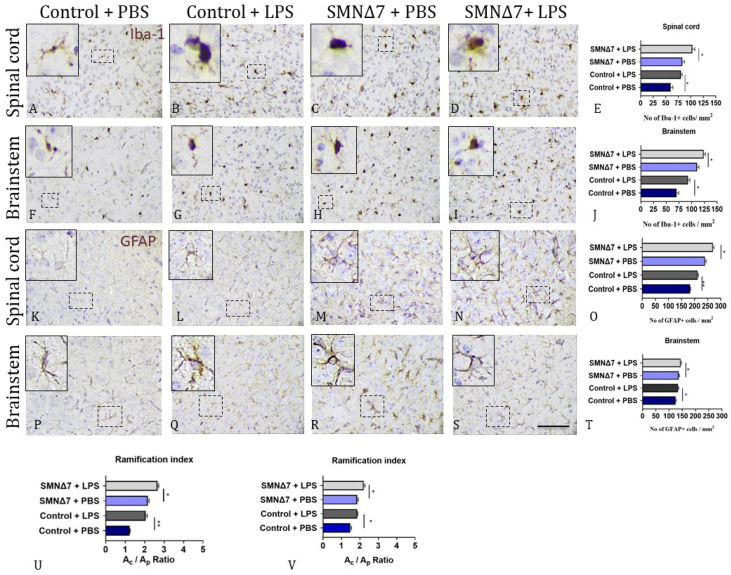
Immunohistochemistry results for microglia and astrocyte populations in CNS with quantitative and morphological changes in microglia and astrocyte dynamics. (**A**–**D**) Iba-1+ microglial cells in spinal cords across experimental groups. (**E**) Graphical illustration of Iba-1+ microglial cells in spinal cords across experimental groups. (**F**–**I**) Iba-1^+^ microglial cells in brainstems across experimental groups. (**J**) Graphical illustration of microglial cells in brainstems across experimental groups. (**K**–**N**) GFAP+ astrocytes in spinal cords across experimental groups. (**O**) Graphical representation of astrocyte counts in spinal cords across experimental groups. (**P**–**S**) GFAP+ astrocytes in brainstems across experimental groups. (**T**) Graphical representation of GFAP+ astrocytes in brainstems across experimental groups. (**U**) Graphical illustration of microglial ramification index in spinal cords across experimental groups. (**V**) Graphical illustration of microglial ramification index in brainstems across experimental groups. Microglial ramification index is used as a morphometric analysis index to detect indications of microglia response/activation to LPS stimulus, and is shown as a ratio of cell body area (Ac) to projection area (Ap) of microglial cells. Regarding the PBS-treated control group, microglial cells presented smaller cell body areas (Ac) divided into wider/bigger projection areas (Ap). This corresponds to small ramification index. Instead, for the LPS-treated mice, the reverse phenomenon was observed. LPS-treated groups demonstrated larger cell body areas corresponding to increased Ac values divided into small projection areas (Ap), since reduced ramification was observed in these cells, leading to an increased ramification index. Experimental groups: control + PBS (dark blue), control + LPS (dark grey), SMNΔ7 + PBS (light blue), and SMNΔ7 + LPS (light gray). Frames indicate inserts. Bars represent mean ± standard error of mean. Error bars represent standard error of mean. Graphs show representative results of two independent experiments, with *n* = 3 for Control + PBS group, N = 5 for Control + LPS group, *n* = 4 for SMNΔ7 + PBS group, and *n* = 5 for SMNΔ7 + LPS group. Ac, cell body area; Ap, projection area; * *p* < 0.05; ** *p* < 0.005; scale: 40×.

**Figure 6 cells-13-00785-f006:**
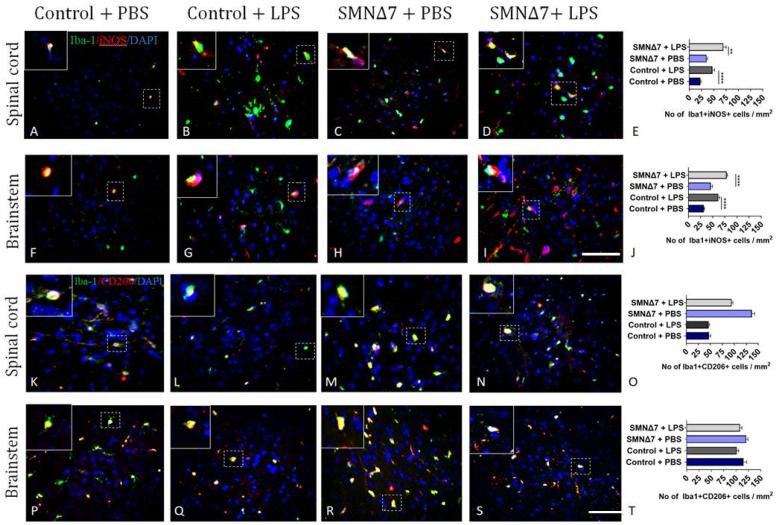
Results of double immunofluorescence staining of microglial cells in two CNS regions reveal activation of different Iba-1+ sub-populations. (**A**–**D**) Results for pro-inflammatory Iba-1+iNOS+ microglia in spinal cord across experimental groups. (**E**) Graphical representation of pro-inflammatory Iba-1+iNOS+ microglial cells in spinal cords across experimental groups. (**F**–**I**) Results for pro-inflammatory Iba-1+iNOS+ microglia in brainstems across experimental groups. (**J**) Graphical representation of pro-inflammatory Iba-1+iNOS+ microglial cells in brainstems across experimental groups. (**K**–**N**) Results for anti-inflammatory Iba-1+CD206+ microglia in spinal cords across experimental groups. (**O**) Graphical representation of for anti-inflammatory Iba-1+CD206+ microglial cells in spinal cords across experimental groups. (**P**–**S**) Results for anti-inflammatory Iba-1+CD206+ microglia in brainstems across experimental groups. (**T**) Graphical representation of anti-inflammatory Iba-1+CD206+ microglial cells in brainstems across experimental groups. Experimental groups: control + PBS (dark blue), control + LPS (dark grey), SMNΔ7 + PBS (light blue), and SMNΔ7 + LPS (light gray). Frames indicate inserts. Bars represent mean ± standard error of the mean. Error bars represent standard error of the mean. Graphs show representative results of two independent experiments, with *n* = 3 for Control + PBS group, *n* = 5 for Control + LPS group, *n* = 4 for SMNΔ7 + PBS group, and *n* = 5 for SMNΔ7 + LPS group. ** *p* < 0.005; **** *p* < 0.0001; scale: 40×.

**Figure 7 cells-13-00785-f007:**
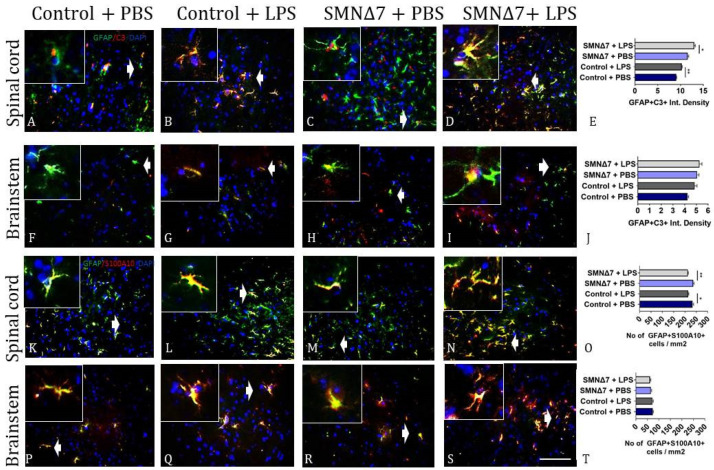
Results of double immunofluorescence staining of astrocytes in two CNS regions reveal activation of different GFAP+ sub-populations. (**A**–**D**) Results for pro-inflammatory GFAP+C3+ astrocytes in spinal cords across experimental groups. (**E**) Graphical representation of pro-inflammatory GFAP+C3+ astrocytes in spinal cords across experimental groups. (**F**–**I**) Results for pro-inflammatory GFAP+C3+ astrocytes in brainstems across experimental groups. (**J**) Graphical representation of pro-inflammatory GFAP+C3+ astrocytes in brainstems across experimental groups. (**K**–**N**) Results for anti-inflammatory GFAP+S100A10+ astrocytes in spinal cords across experimental groups. (**O**) Graphical representation of anti-inflammatory GFAP+S100A10+ astrocytes in spinal cords across experimental groups. (**P**–**S**) Results for anti-inflammatory GFAP+S100A10+ astrocytes in brainstems across experimental groups. (**T**) Graphical representation of anti-inflammatory GFAP+S100A10+ astrocytes in brainstems across experimental groups. Experimental groups: control + PBS (dark blue), control + LPS (dark grey), SMNΔ7 + PBS (light blue), and SMNΔ7 + LPS (light gray). Frames indicate inserts. Bars represent mean ± standard error of the mean. Error bars represent standard error of the mean. Graphs show representative results of two independent experiments, with *n* = 3 for Control + PBS group, *n* = 5 for Control + LPS group, *n* = 4 for SMNΔ7 + PBS group, and *n* = 5 for SMNΔ7 + LPS group. * *p* < 0.05; ** *p* < 0.005; scale: 40×.

## Data Availability

The data presented in this study are available upon request from the corresponding author.

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
