# Peer review of "Systemic LPS Administration Stimulates the Activation of Non-Neuronal Cells in an Experimental Model of Spinal Muscular Atrophy"

_cells, 2024, doi:10.3390/cells13090785_

Round 1

Reviewer 1 Report

Comments and Suggestions for Authors

SMN protein, encoded by the SMN1 and SMN2 genes, is expressed in all animal cells, and it plays important roles in multiple fundamental cellular homeostatic pathways. A mutation in the SMN1 gene causes a deficiency of SMN protein leading to spinal muscular atrophy (SMA) because of the selective death of motor neurons, a key feature of the SMA disease. To examine the impact of bacterial infection on the pathogenesis of SMA, this study intended to explore the effect of administration of LPS to control and SMNd7 mice on some motor functions and the expression of SMN in the brainstem, spinal cord, some muscle, as well as some peripheral tissues.

Below are this reviewer’s major and minor comments on issues in the following sections of this manuscript.

Methods:

-          Major comment: 5μg Ε.coli LPS was administered by intra-peritoneal injection on neonates at postnatal day 5 (P5). Although 5μg is a sub-lethal dose, it is a very high dose of LPS for postnatal day 5 mice, in particular for those SMNd7 mice. This issue needs to be discussed in the context of relevant results.

Figure 1:

-          Minor comment: Usually the labels of A, B, C, D are placed on the top of subfigures. 

-          Major comment: The graphic traces of LPS effects on motor functional scores in Figures 1C and 1D cannot be seen – Are they overlaid by the traces of PBS treatment? If this is the case, the authors should state the fact in the figure legend, or show these traces in separate sub-figures.

-          Major comment: The figure legend states: “Graphs show representative results of two independent experiments with 3-5 mice per experimental group”. The numbers of mice used in each experimental groups seem too small for drawing a solid conclusion.  

Figure 2:

-          Major comment: To support the statement “Systemic LPS administration promoted a reduction of SMN expression levels in CNS, peripheral lymphoid organs and skeletal muscles. Moreover, peripheral tissues were more vulnerable to LPS-induced damage compared to CNS tissues”, the authors should statistically examine differences in the ratio of SMN protein expression before and after the LPS treatment, between CNS tissues and peripheral organs.

-          Minor comment: Please specify “brain” tissues – were they tissues of the whole-brain, brainstem, or specific brain region(s) of test mice?  

-          Major comment: Figure 2G shows no difference in levels of SMN protein expression in the spleen among animal groups. Authors should discuss this result in the context that the size of the spleen in SMND7 mice is significantly smaller than that in control mice. Also SMND7 mice display alterations in lymphocyte concentration and macrophage population (Human Molecular Genetics, Volume 26, Issue 5, 1 March 2017, Pages 932–941).

-          Major comment: Although LPS treatment significantly decreased the expression level of SMN proteins in the central and peripheral tissues, it did not affect the motor functions in control and SMNd7 mice (Figure 1). Together, these results are inconsistent with the authors’ statement that “systemic LPS administration may precipitate phenotype alterations in a mouse model of SMA…” because the main “phenotype alterations” in SMA are deficits of motor functions.

Figure 3:

-          Major comment: The numbers of tissue sections and the numbers of mice in each group should be indicated in the text of result descriptions or in the figure legend.

-          Minor comment: Regarding of analyses of Nissl+ motor neurons, as well as Bielschowsky staining in the brainstem, the region(s)/nucleus of the brainstem of mice shown in the figures should be indicated.

-       Minor comment: The figure legend should indicate each of the subfigures 3P, Q, R, and S. If Figure 3S represents SMNd7+LPS, it looks not “matching” the score of SMNd7+LPS in Figure 3T.

Figure 4:

-          Major comment: Regarding immunostaining of synaptophysin in the brainstem, the region/nucleus should be indicated.

-          Minor comment: In comparison to spinal motor neurons (shown in Figures 4A-D), the size of cell bodies of neurons illustrated in Figures 4H and F looks too small to represent motor neurons in the brainstem.

-          Major comment: The density of synaptophysin immunofluorescence surrounding motor neurons represents the quantity of synaptic terminals from premotor neurons including spinal interneurons and sensory neurons. The authors may discuss the LPS-induced decrease of synaptophysin in the context of motor neuron degeneration of SMA.

Figures 5, 6 and 7:

-          Major comment: The numbers of tissue slices and the number of animals in each group should be indicated in the text of relevant results or in figure legends.  

Discussion:

-          Major comment: In different paragraphs of Discussion, the authors made following statements: 1. “LPS-mediated modifications of alternative splicing and gene transcription were previously shown to lead in a decrease of SMN protein levels in the CNS and peripheral tissues [33–38]”. 2. “LPS-induced neurodegeneration and synaptic damage has been established previously in vitro [54] and this effect was shown to be microglial dependent and associated with microglia-derived interleukin-1beta (IL1β) or tumor necrosis factor alpha (TNF-a) levels [54,55]. 3. “Further insights into the effect of innate immunity cells in neurodegenerative diseases, including SMA, have also linked glial cells [56–59] with neuroinflammation leading to neurodegeneration [60, 61]. Existing evidence underlines the implication of astrocytes and microglia in SMA pathogenesis [57], and these populations are susceptible to the effect of systemic microbial stimuli [62]”. However, in the last paragraph of Discussion, the authors concluded: “Our study provides for the first time thorough mechanistic insight on the effect of systemic administration of microbial LPS on experimental SMA in terms of SMN expression, clinical disease outcomes, as well as the equilibrium between neurodegeneration and the activation of innate immunity in the CNS”. This reviewer feels that these statements are inconsistent.

Reviewer 2 Report

Comments and Suggestions for Authors

This is an important study that might further support some of the anecdotal reports of SMA symptom onset in infants after illness, particularly gastrointestinal infections. 

General comments:

To do justice to the very nice data, sections of the manuscript (particularly the results) should be rewritten to improve clarity and readability.  Some examples of how readability might be improved are provided below. 

The animals used in this study are SMND7 and control mice- use the term ‘mice/mouse’ not ‘rodents’

Specific comments:

Introduction;

Line 37-38 Rephrase the first sentence- Spinal muscular atrophy (SMA) belongs to polysystemic ‘orphan’ diseases and is a  consequence of endogenous default of the survival motor neuron (SMN) protein.

SMA is a consequence of homozygous loss of SMN1 expression and insufficient expression of functional SMN from the SMN2 gene.

Line 40:- correct a-MN to α-MNs

Methods:

Line 101, 103: change ‘lane on ‘ to ‘product of’

Line 108: change ‘control’ to ‘vehicle.  The term ‘control’ is also used for the isogenic control mice.

Line 122: perhaps delete ‘a customized device consisting of’- it is just a plastic centrifuge tube.

Line 116: delete ‘scale’.  In the tail suspension test, the scores are described from 0 to 4. The next test (Hind limb) reverses the order, beginning with 4. Readability can be improved by always presenting information/data in the same order.

Line 123: The following should be rephrased. 

A cotton cushion was used in order to avoid pups’ injury in case of fall.

‘Pups were protected from injury in case of fall by placement of a cotton cushion. 

Line 141: 20-30 micro gm is regarded as an excessive amount of total tissue protein to load on a gel, and likely exceeds the unit binding capacity of the membrane.  The western blot images show very strong signals, particularly for the housekeeper proteins, very likely to lie outside the dynamic range of the detection system and compromise the quantitation.

Results:

Many sentences are convoluted and much too long (>60 words); begin with ‘Concerning…’ or ‘Regarding…’ ‘With respect to…’.  Readability could be improved by splitting sentences and maintaining the subject order throughout.

For example;

Line 242-245: With respect to the tail suspension test (Figure 1 C) the same trend was found in mice for both PBS-injected and LPS-injected groups of control (mean score 3.94 ± 0.03 vs 3.92 ± 0.05, p>0.99) and SMNΔ7 (mean score 1.78 ± 0.49 vs 1.78 ± 0.49, p>0.99) and results showed that they remained unaffected by LPS-administration and exhibited no additional  neuromuscular burden.

 Can be rephrased-

Administration of LPS did not affect the performance of the SMNΔ7 or the control mice in the tail suspension test.   The mean scores of the SMNΔ7 LPS- and PBS-treated mice on this test  were 1.78 ± 0.49 vs 1.78 ± 0.49 (p>0.99), respectively, while the scores for the LPS- and PBS- treated control mice were 3.94 ± 0.03 vs 3.92 ± 0.05 (p>0.99), respectively.

Also

Line 302-306:

To further examine the number of neurons, we performed Nissl staining on spinal cord (Figure 3 A-E) no statistically significant difference was found following LPS administration, neither in the control group (73.2 ± 6.93 vs. 77.1 ± 2.98, p>0.99, with vs. without LPS administration, respectively), nor in the SMNΔ7 mice (33.2 ± 4.06 vs. 33.7 ± 2.65, p>0.99, with vs. without LPS administration, respectively).”

Could be rephrased

‘We performed Nissl staining on spinal cord to assess neuron number and determined that LPS administration did not result in a significant difference in motor neuron number in either the SMNΔ7 or control groups (Figure 3 A-E), however, the control spinal cords had more than twice as many motor neurons per unit area.  SMNΔ7 mice treated with LPS or PBS had 33.2 ± 4.06 and 33.7 ± 2.65 motor neurons/ mm2 , respectively (p>0.99), while the LPS treated control mice had 73.2 ± 6.93 and those treated with PBS had 77.1 ± 2.98 motor neurons/ mm2 (p>0.99).’

Figures: All Figure legends require additional detail so that the data can be interpreted without having to return to the text.

Figure 1:  The graphs are not interpretable, for C and D only 2 lines are evident.  Improve resolution, enlarge or use different symbols so that the data points are visible.

Figure 2: The graphs do not reflect the data on the western blot images.  The brain SMN expression in control cohorts, as shown on the blot is at least 10-20 fold that in the SMNdelta7 mouse brains.  The graph show ~2-fold different.  Beta actin is not the ideal housekeeper for a relatively low- expression protein, its expression is too high and likely exceeds the binding capacity of the membrane and the linear signal range.  Ideally, the control samples should be loaded over a range- eg 5, 10, 15 micro gm for better comparison the SMNdelta7 samples.  The legend also needs to explain that the expression level in the control PBS-treated sample was set at 1.0. Sample number is not stated.

Figure 3:The ‘mean score’ in the graphs needs to be explained in the legend.

Figure 4: expalain ‘Intl. density of SYP’

Figure 5: explain ‘microglial ramification index’ in the context of these graphs.

Figure 6 and 7: the graphs are too small.

Discussion:

Line 503: replace ; ‘crossroad’ with ‘intersection’

Line 519: insert ‘mice after ‘SMNΔ7’

Line 572: insert ‘transcript’ after ‘SMN2 gene’

The novelty of the study is perhaps a little overstated towards the end of the discussion.  It would be worthwhile to emphasise that LPS challenge in SMA mice has been reported before (ref 33) the current study explores the CNS impact in greater detail and in a different (less severe) mouse model.

Comments on the Quality of English Language

As above, the language is often convoluted and many statements are difficult to interpret, requiring unreasonable effort from the reader.  The report is nicely structured and presents some very nice data that will be of much interest to the field.  Guidance from a native English scientific writer is recommended.

Round 2

Reviewer 1 Report

Comments and Suggestions for Authors

Authors addressed my major concerns.

Reviewer 2 Report

Comments and Suggestions for Authors

The authors are to be commended for the revised manuscript, and the careful attention to the reviewer's suggestions and for including the additional detail as requested. 

Comments on the Quality of English Language

Some minor editing and correction of grammar and spelling errors are required: eg

Line 527-34: throughout Figure 5 legend rephrase 'LPS stimulus and it is a ratio of cell body area (Ac) divided to projection area (Ap) of microglial cells ' .....change 'to' to 'by

Line 846-8: rephrase '.....LPS challenge in SMA models is that addresses CNS impact more thoroughly and in a less severe ....'